# Incident mobility disability, parkinsonism, and mortality in community-dwelling older adults

**Shahram Oveisgharan** [1,2]*, **Lei Yu**[1,2], **David A. Bennett**[1,2], **Aron S. Buchman**[1,2]

**1** Rush Alzheimer's Disease Center, Rush University Medical Center, Chicago, IL, United States of America,
**2** Department of Neurological Sciences, Rush University Medical Center, Chicago, IL, United States of America

* shahram_oveisgharan@rush.edu

**Data Availability Statement:** This study used data from the Rush Memory and Aging Project (R01AG17917) and the Rush Religious Orders Study (P30AG10161) obtained from the Rush Alzheimer's Disease Center (RADC) repository. The

## Abstract

### Background

Mobility disability and parkinsonism are associated with decreased survival in older adults. This study examined the transition from no motor impairment to mobility disability and parkinsonism and their associations with death.

### Methods

867 community-dwelling older adults without mobility disability or parkinsonism at baseline were examined annually. Mobility disability was based on annual measured gait speed. Parkinsonism was based on the annual assessment of 26 items from the motor portion of the Unified Parkinson's Disease Rating Scale. A multistate Cox model simultaneously examined the incidences of mobility disability and parkinsonism and their associations with death.

### Results

Average age at baseline was 75 years old and 318 (37%) died during 10 years of follow-up. Mobility disability was almost 2-fold more common than parkinsonism. Some participants developed mobility disability alone (42%), or parkinsonism alone (5%), while many developed both (41%). Individuals with mobility disability or parkinsonism alone had an increased risk of death, but their risk was less than the risk in individuals with both impairments. The risk of death did not depend on the order in which impairments occurred.

### Conclusion

The varied patterns of transitions from no motor impairment to motor impairment highlights the heterogeneity of late-life motor impairment and its contribution to survival. Further studies are needed to elucidate the underlying biology of these different transitions and how they might impact survival.

repository consent form signed by participants was approved and is overseen by the local Rush IRB. All of the data is available, but is governed by the data sharing plan for publicly sharing the RADC data repository that was approved by the local RUSH IRB and for which the participants consented. This plan was also approved by the primary funding agency, the NIA. The data underlying the results presented in this study can be requested at http://www.radc.rush.edu.

**Funding:** This work was supported by National Institute of Health grants, R01AG043379, R01AG047679, R01AG056352, R01AG017917, RF1AG022018, R01NS078009, P30AG10161, and R01AG15819. We embrace that "The funders had no role in study design, data collection and analysis, decision to publish, or preparation of the manuscript." In addition, authors received salary through the grants supported by National Institute of Health.

**Competing interests:** The authors have declared that no competing interests exist.

## Introduction

Motor impairment is common and affects up to half or more of older adults [1–3] and is associated with diverse adverse health outcomes including death [4–6]. Impaired motor function in older adults is heterogeneous ranging from symptoms of poor strength, slowed walking, imbalance, reduced dexterity to motor disability. Investigators have developed a wide variety of clinical instruments that target the assessment of different motor abilities when testing motor function of older adults. For example, handgrip strength and gait speed performances are important features of several measures including sarcopenia and physical frailty. By contrast, parkinsonism focuses on other motor signs and symptoms, parkinsonian gait, tremor, rigidity and bradykinesia, which commonly manifest in older adults.

Many studies suggest that impairment of different motor phenotypes is associated with reduced survival, but few studies have examined the inter-relationship among multiple motor phenotypes and their association with survival in the same individuals [6–8]. Previously, we reported that gait speed and the severity of parkinsonism in the same individuals are independently related to risk of mortality [9]. These findings suggest that these two phenotypes are not duplicative, but rather may capture distinct facets of motor function that might explain why their combination is more strongly associated with risk of death than either alone. This prior study examined the association of continuous measures of these two motor phenotypes at one point in time with survival. Thus, it is unknown whether the risk of death varies as older adults without initial mobility disability or parkinsonism develop one or both of these motor impairments over time or if the order of the development of these impairments affects an individual's risk of death. These knowledge gaps impede risk stratification of older adults and public health intervention efforts to reduce these common motor phenotypes and improve survival.

To fill these knowledge gaps, the current study employed a novel analytic method, used in some recent studies [10], to examine the relationship of mobility disability and parkinsonism and their relation to death. Specifically, we had four objectives. First, we estimated the incidence of the two phenotypes. Second, we tested the hypothesis that mobility disability increases the risk for parkinsonism and vice versa. Third, we examined whether incident mobility disability and incident parkinsonism, alone or together, was associated with the risk of death. Fourth, we examined whether the sequence of the occurrence of these disabilities was associated with the risk of death. We analyzed data from 867 community-dwelling older adults, participating in one of two community-based longitudinal studies, who were initially without mobility disability and parkinsonism and underwent repeated annual assessments.

## Methods

### Participants

The Religious Orders Study (ROS) began recruitment in 1994. The Rush Memory and Aging Project (MAP) began recruitment in 1997. Eligible participants in both studies were adults older than 65 years without known dementia, and both studies are ongoing. ROS recruits nuns, priests, and brothers across the United States. MAP recruits participants living in private homes, subsidized housings, and retirement facilities across the greater Chicago metropolitan area. Both studies employ harmonized data collection methods performed by the same staff, including participants' consent to annual testing during life and to Anatomical Gift Act at the time of death. Harmonized data collection facilitates joint analyses of the studies' data. Details of the studies are described elsewhere [11]. A Rush University Medical Center Institutional Review Board approved each study. Data from these studies can be obtained via requests uploaded to www.radc.rush.edu.

**Table 1. Baseline clinical characteristics based on final pattern of motor impairments developed during the study.**

| Covariates | All (n = 867) | NI (n = 110)[‡] | MD (n = 362)[‡] | PARK (n = 40)[‡] | Park\|MD (n = 236)[‡] | MD\|PARK (n = 119)[‡] |
|---|---|---|---|---|---|---|
| **Demographics** | | | | | | |
| Age, years, mean (SD) | 75.5 (7.4)** | 74.1 (7.3) | 73.6 (7.5) | 78.7 (6.6) | 76.9 (7.0) | 78.4 (6.5) |
| Female, n (%) | 658 (76) | 76 (69) | 281 (78) | 27 (68) | 189 (80) | 85 (71) |
| Education, years, mean (SD) | 16.5 (3.5)* | 17.1 (3.6) | 16.6 (3.4) | 15.7 (3.3) | 16.4 (3.5) | 15.7 (3.4) |
| **Chronic health conditions** | | | | | | |
| **Vascular risk factors and diseases** | | | | | | |
| Hypertension, n (%) | 403 (47) | 52 (47) | 166 (46) | 20 (50) | 113 (48) | 52 (44) |
| Diabetes, n (%) | 87 (10)* | 10 (9) | 38 (11) | 10 (25) | 16 (7) | 13 (11) |
| History of smoking, n (%) | 297 (34) | 41 (37) | 128 (35) | 13 (33) | 77 (33) | 38 (32) |
| Number of 3 vascular risk factors, mean (SD) | 0.9 (0.8) | 0.9 (0.8) | 0.9 (0.8) | 1.1 (0.9) | 0.9 (0.8) | 0.9 (0.8) |
| History of stroke, n (%) | 43 (5) | 5 (5) | 16 (5) | 2 (5) | 13 (6) | 7 (6) |
| History of myocardial infarction, n (%) | 63 (7) | 6 (5) | 26 (7) | 5 (13) | 16 (7) | 10 (8) |
| History of claudication, n (%) | 40 (5) | 1 (1) | 18 (5) | 1 (3) | 15 (6) | 5 (4) |
| Number of 3 vascular diseases, mean (SD) | 0.2 (0.4) | 0.1 (0.3) | 0.2 (0.4) | 0.2 (0.4) | 0.2 (0.4) | 0.2 (0.5) |
| **Musculoskeletal pain (any joint), n (%)** | 331 (38) | 33 (30) | 133 (37) | 13 (33) | 107 (45) | 45 (38) |

[‡]**NI**: Remained without mobility impairment and parkinsonism during the study; **MD**: Developed only mobility disability; **PARK**: Developed only parkinsonism;

**PARK|MD**: Developed parkinsonism following mobility disability; **MD|PARK**: Developed mobility disability following parkinsonism.

** $P< 0.001$.

* $P< 0.05$.

The analytic sample for the current study included participants without mobility disability and parkinsonism at baseline. As cognitive and motor impairment are related [12], we excluded participants with cognitive impairment at baseline. Of 1469 ROS participants recruited through October 2019, 1009 were without cognitive impairment at baseline, of whom 464 did not have either mobility disability or parkinsonism assessment and 236 had mobility disability or parkinsonism at baseline. From 309 participants without motor or cognitive impairment at baseline, 19 were lost to follow up that leaving 290 ROS participants for the current study.

Through October, 2019, 1484 of 2164 MAP participants were without cognitive impairment at baseline, 464 did not have either mobility disability or parkinsonism assessment and 236 had mobility disability or parkinsonism at baseline. Of 659 MAP participants without motor or cognitive impairment at baseline, 82 were lost to follow up leaving 577 adults for these analyses. The clinical characteristics for the 867 adults included in this study [MAP (577); ROS (290)] are summarized in **Table 1**.

## Assessment of motor impairment

**Assessment of mobility disability.** Annual testing included a self-paced 8ft walking task. Although self-reported questionnaires are available for assessment of mobility disability, studies have shown objective measures like gait speed assessment to be a more sensitive and accurate method for identifying mobility disability [13, 14]. As previously published, incident mobility disability in these analyses was defined as the first visit at which measured walking speed was less than 0.55 m/s in the 8ft walk [15, 16].

**Assessment of parkinsonism.** Nurse clinicians assessed parkinsonian gait, rigidity, bradykinesia, and tremor annually using 26 items from a modified motor section of the original United Parkinson's Disease Rating Scale (UPDRS) [17]. These measures have high inter-rater

reliability and short-term stability among nurses and compared with a movement disorders specialist [18].

Four parkinsonian signs including parkinsonian gait, bradykinesia, rigidity, and tremor were assessed as described in prior publications [19–21]. A parkinsonian sign was present if two or more of the items assessed for that sign showed a mild abnormality. Incident parkinsonism was defined as the first visit at which two or more of the four parkinsonian signs were present. Clinical diagnosis of Parkinson's disease (PD) was based on self-reported diagnosis of PD for which the participant received L-dopa or a dopamine agonist [21, 22].

## Assessment of other clinical variables

Age was computed from self-report date of birth and date of baseline clinical assessment. Sex and years of education were recorded at study baseline. Self-report data at baseline also provided information about participants' history of hypertension, diabetes, smoking, heart attack, stroke, lower extremities claudication, and joint pain.

The annual structured evaluation included administration of a battery of 17 cognitive tests. A neuropsychologist reviewed the cognitive tests' results, and a physician with expertise in dementia reviewed annual findings including the summary of the neuropsychologist and classified the cognitive status of participants as no cognitive impairment, mild cognitive impairment, and dementia including probable and possible Alzheimer's disease, according to established criteria [23]. In this study, we excluded participants whose cognitive status at the baseline was mild cognitive impairment (n = 895) or dementia (n = 212), or the cognitive status could not get determined (n = 33).

## Statistical analyses

We employed a multi-state Cox model with six states to model the transition from no motor impairment to incident mobility disability and/or incident parkinsonism and finally to death [10]. All participants began with no motor impairment (NMI) at study baseline. The second to fifth states were intermediate states including: 2) incident mobility disability alone (MD); 3) incident parkinsonism alone (Park); 4) incident parkinsonism following mobility disability

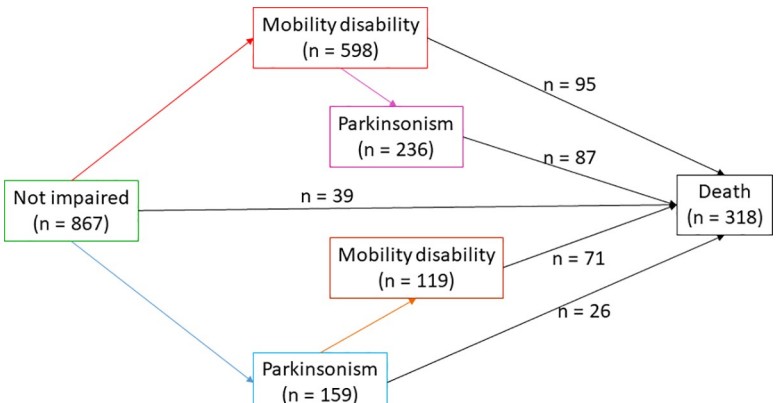

**Fig 1. States and transitions in a multi-state model of incident mobility disability, incident parkinsonism, and risk of death.** This figure illustrates the frequencies of participants in each of the six states and nine transitions examined in a multi-state model of incident mobility disability, parkinsonism, and death. All participants included in this study did not show mobility disability or parkinsonism at the analytic baseline. During the course of this study, nine paths for transition were possible between the baseline state of no motor impairment, four intermediate states of varying degrees of motor impairment, and the final absorbing state of death.

(Park|MD); 5) incident mobility disability following parkinsonism (MD|Park). The sixth state was the final absorbing state of death (**Fig 1**). The model structure included 9 distinct transitions as follows:

$$\begin{bmatrix}
 & NMI & MD & Park & Park|MD & MD|Park & Death \\
NMI & - & \lambda_{12}(t|Z) & \lambda_{13}(t|Z) & - & - & \lambda_{16}(t|Z) \\
MD & - & - & - & \lambda_{24}(t|Z) & - & \lambda_{26}(t|Z) \\
Park & - & - & - & - & \lambda_{35}(t|Z) & \lambda_{36}(t|Z) \\
Park|MD & - & - & - & - & - & \lambda_{46}(t|Z) \\
MD|Park & - & - & - & - & - & \lambda_{56}(t|Z)
\end{bmatrix}$$

$\lambda_{ij}(t|Z)$ is an estimate of the hazard of transition from state i to state j, and is modelled as the following:

$$\lambda_{ij}(t|Z) = \lambda_{ij,0}(t)\exp(\beta_{ij}^{T}Z)$$

The $\lambda_{ij,0}$ is the baseline hazard, Z is the vector of covariates (including demographics and clinical variables), and β quantifies the association of Z with the hazard. In this study, we had nine hazard functions corresponding to the nine possible transitions. The model allows the associations of Z to be transition-specific.

The main objective was to determine whether the sequence of occurrence of mobility disability and parkinsonism was associated with the risk of death. To accomplish this objective, we assumed proportional hazards of $\lambda_{46}$ and $\lambda_{56}$ as follows,

$$\lambda_{56}(t|Z) = \lambda_{46,0}(t)\exp(\beta_{i6}^{T}Z + \delta)$$

The indicator δ tested the difference in the baseline hazard of death between MD|Park (i.e. mobility disability after Parkinsonism) and Park|MD (i.e. Parkinsonism after mobility disability). The proportional assumption was checked by using Schoenfeld residuals against the transformed time [24].

We took similar approach to examine whether the risk of impairment in either of the motor phenotypes was higher if impairment of the other phenotype had already occurred. We also examined whether risk of death was higher if either mobility disability alone, parkinsonism alone, or both impairments had developed compared to no prior motor impairment. The analyses were done using SAS/STAT software, version 9.4 (SAS Institute, Cary, NC, USA), and mstate package for R [25]. P values less than 0.05 was required to reject the null hypotheses.

## Results

The 867 adults included in this study were on average 75 years old, and three fourth of them were women. During an average follow-up of 10 years (mean 10.2 yrs, SD = 5.2, range: 1–22), 318 (37%) died. As illustrated in **Fig 1**, individuals were classified into five different categories based on the presence or absence of motor impairment before death or the end of follow up. 1) Some participants remained without motor impairment (N = 110). Others transitioned into one of four states of motor impairment including: 2) mobility disability only (N = 362), 3) parkinsonism only (N = 40), 4) mobility disability and subsequently parkinsonism (N = 236), and 5) parkinsonism and subsequently mobility disability (N = 119). The baseline clinical characteristics and demographics for the entire analytic cohort and the five groups based on their final status of motor impairment are included in **Table 1**.

## Incident mobility disability

Mobility disability occurred in 717 (83%) after an average 4.7 years (Fig 1). The risk of developing mobility disability was not different in participants who first developed parkinsonism compared to participants without motor impairment (p = 0.526). During five years of follow-up (Fig 2), an average participant (female 75 years old with 16 years of education) with or without parkinsonism showed an increased risk of developing mobility disability [Risk of mobility disability after 5 years: without prior parkinsonism: 36% (95% CI: 33% - 40%) versus prior parkinsonism: 34% (95% CI: 26% - 41%)].

## Incident parkinsonism

Parkinsonism occurred in about half (N = 395, 46%) of the participants after an average 5.3 years of follow up (Fig 1). Compared to participants without motor impairment, the risk of developing parkinsonism was higher in participants who first developed mobility disability (HR = 3.1, 95%CI: 2.5–4.0, p<0.001). This association persisted after controlling for age, sex, and education (HR = 2.6, 95%CI: 2.0–3.3, p<0.001).

Fig 3 illustrates the increased risk of developing parkinsonism in two average participants (female 75 years old with 16 years of education) with and without prior mobility disability. Over five years, the risk of developing parkinsonism is nearly 4-fold higher for a participant who first developed mobility disability, 39% (95% CI: 31% - 46%), compared to a participant without motor impairment 10% (95% CI: 8% - 12%).

## Incident motor impairment and risk of death

Next we examined if the risk of death associated with incident mobility disability and incident parkinsonism were different. First, we examined whether the sequence of the occurrence of

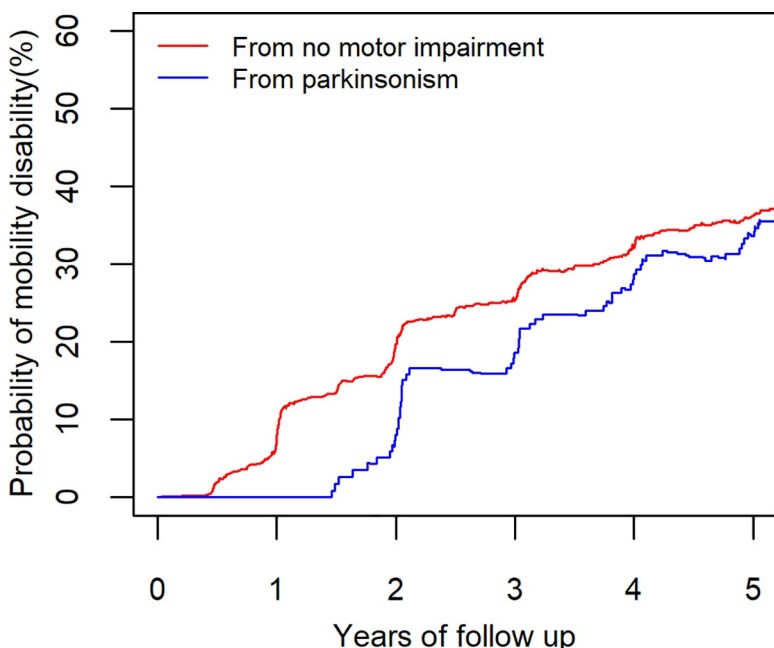

**Fig 2. Probability of developing mobility disability with and without prior parkinsonism.** After five years, two average participants (female 75 years old with 16 years of education) without parkinsonism (red lines) and with parkinsonism (blue lines) show a similar risk for developing mobility disability of about 35%.

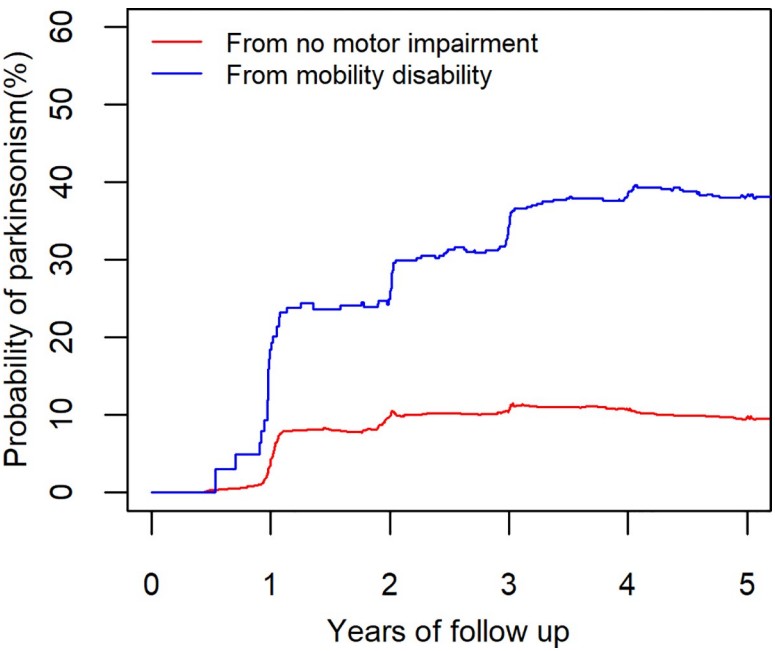

**Fig 3. Probability of developing parkinsonism in individuals with and without prior mobility disability.** The risk of developing parkinsonism in two average participants' (female 75 years old and 16 years of education) without (red line) mobility disability and with prior mobility disability (blue line). The participant with mobility disability had a higher risk.

mobility disability and parkinsonism was associated with a different risk of death. The risk of death was not different in participants who developed mobility disability after first developing parkinsonism compared to participants who developed parkinsonism after first developing mobility disability (HR = 1.2, 95%CI: 0.9–1.6, p = 0.318).

Next, we compared the risk of death in the 3 groups of individuals who developed motor impairment prior to death (mobility disability alone, parkinsonism alone, and both mobility disability and parkinsonism) with a reference group of participants who did not develop motor impairment during the study. The risk of death was higher for all 3 groups of individuals who had developed motor impairment. For participants with incident mobility disability alone, the HR of death was 1.8 (95%CI: 1.2–2.6, p = 0.004). For participants with incident parkinsonism alone, the HR was 2.8 (95%CI: 1.7–4.7, p<0.001). For participants with both mobility disability and parkinsonism prior to death, the HR was 4.0 (95%CI: 2.7–5.9, p<0.001).

After controlling for demographics, incident motor impairment remained associated with the risk of death, but the association was attenuated: [incident mobility disability alone (HR = 1.5, 95%CI: 0.99–2.2, p = 0.059), incident parkinsonism alone (HR = 2.1, 95%CI: 1.3–3.5, p = 0.003), incident mobility disability and parkinsonism (HR = 2.5, 95%CI: 1.7–3.7, p<0.001)].

**Fig 4** illustrates the estimated probability of death for four average participants (75 years old woman with 16 years of education) without and with different patterns of motor impairments prior to death. During 10 years of follow-up, the risk of death observed in individuals with both mobility and parkinsonism was almost two-fold higher compared to individuals without motor impairment. Risk of death was increased but intermediate for individuals with either mobility disability alone or parkinsonism alone.

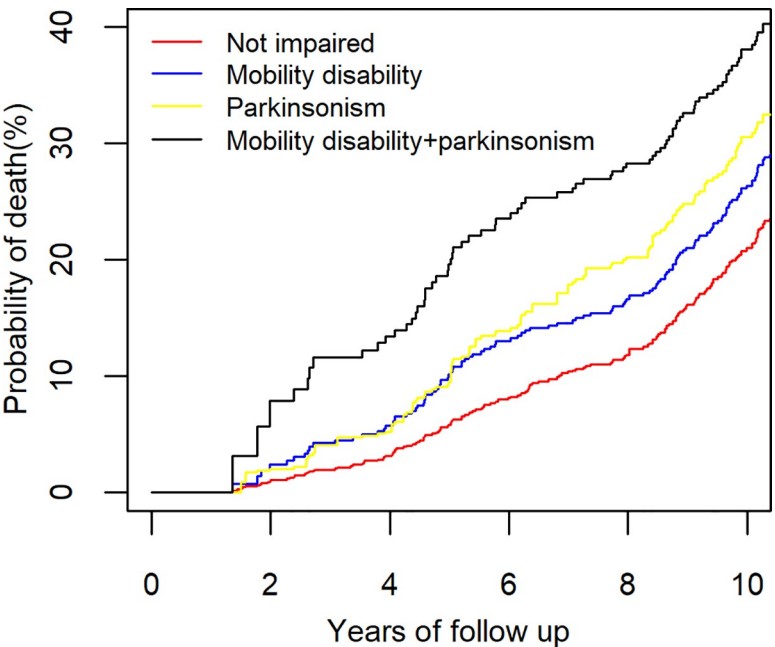

**Fig 4. Probability of death in older adults with and without prior motor impairments.** This figure illustrates the risk of death after ten years for four average participants (Female 75 years old with 16 years of education) with and without different patterns of motor impairments. Risk of death in the individual with both mobility disability and parkinsonism (black line, 38%) was nearly 2-fold higher than an individual without any motor impairment (red line, 21%). Individuals with only mobility disability (blue, 26%) or only parkinsonism (grey, 31%) showed an intermediate increased risk of death compared to individuals with both impairments or no motor impairment.

## Secondary analyses

We examined whether controlling for chronic health conditions attenuated the associations of incident motor impairments and the risk of death (**S1–S3 Tables**). Vascular risk factors were not associated with any of the 9 transitions. Vascular diseases increased the risk of death in participants who developed only mobility disability. Joint pain increased the risk of mobility disability in participants without any motor impairment. However, further adjustment for vascular diseases or joint pain did not change the inter-relationship of incident mobility disability and incident parkinsonism or their associations with risk of death (**S4 and S5 Tables**).

In further analyses, we explored whether inadequate power was responsible for the finding that incident mobility disability showed a trend for an association with death. The examined model had nine possible transitions compared to only two transitions in previous studies (Yes/No mobility disability and death) [26]. In a sensitivity analyses, we reduced the number of states and transitions, which increased the sample size per transition. In this model, the first state was the baseline state of NMI. The second, intermediate, state was incident mobility disability (with or without prior parkinsonism), and the final absorbing state was death (**S1 Fig**). As death could occur from baseline or from incident mobility disability, and participants could develop mobility disability during follow up without death, the model structure included 3 distinct transitions (**S1 Fig**, number of arrows). In this model with increased power due to the reduced transitions and states, mobility disability was a risk factor for death, even after controlling for age, sex, and education (HR = 1.5, 95%CI: 1.1–1.9, p = 0.008). The point estimate of the association of mobility disability and death was the same in the reduced (3 transitions) and the full (9 transitions) models, a finding that supports inadequate power to be responsible for the trend of association between mobility disability and death.

Parkinsonism is a heterogeneous syndrome including older adults receiving different medications, diverse medical conditions and degenerative disorders including PD. To ensure that our findings were not affected by a subset of individuals with PD, we excluded individuals with a clinical diagnosis of PD (n = 14) and repeated our analyses. Our primary findings were unchanged: mobility disability was a risk factor for incident parkinsonism and the highest risk of death was among participants with both mobility disability and parkinsonism (**S2 Fig, S6 Table**).

Of the four parkinsonian signs assessed in this study, bradykinesia is considered a cardinal sign for diagnosis of Parkinson disease. Therefore, to examine the robustness of our findings, we repeated our analyses after redefining incident parkinsonism based on the presence of bradykinesia together with at least another parkinsonian sign (tremor, rigidity, parkinsonian gait). Our primary findings showing that mobility disability increases the risk of parkinsonism and that individuals with both mobility disability and parkinsonism have the highest risk of death were unchanged (**S3 Fig, S7 Table**).

## Discussion

This study employed novel modeling to simultaneously examine the transition from no motor impairment to one or both of incident mobility disability and parkinsonism, and whether these transitions were associated with different risks of death. We found that the transition from no motor impairment to motor impairment in older adults is more heterogeneous than currently appreciated. Some individuals may develop mobility disability or parkinsonism alone prior to death. However, these latter impairments represent an intermediate stage in many individuals as they eventually develop both impairments prior to death. A novel feature revealed by the transition state modeling was that both the frequency and the onset of incident mobility disability and incident parkinsonism during the study differed. Mobility disability was almost 2-fold more common than parkinsonism and the latter occurred later during follow-up. We found that both phenotypes are not uniformly associated with risk of death. While, individuals who developed either mobility disability or parkinsonism alone had an increased risk of death, parkinsonism was more strongly associated with death than mobility disability. However, the risk of death was the highest in individuals who developed both impairments but the order of their occurrence did not affect the increased risk of death. Together, these data suggest that mobility disability based on slowed gait speed and parkinsonism are related but distinct motor impairment phenotypes.

Many previous studies suggest that late-life motor impairment is associated with increased risk of adverse health outcomes, including death [5, 6, 8, 17]. However, late-life motor impairment is manifested by different motor phenotypes and most studies have focused on a single motor phenotype. In prior work, we reported that more severe parkinsonism is related to poor mobility in the same individuals and presence of both phenotypes is more strongly related to the risk of death [27]. These data were derived from standard survival models in which both phenotypes were measured at a single point in time, i.e., the analytic baseline [28]. These models cannot be used to assess intermediate state changes of the phenotype of interest or how a phenotype might be affected by other phenotype that may precede or follow its onset. To circumvent these limitations, we applied multistate modeling that was successfully employed in our prior study to examine the onset of incident cognitive impairment and incident mobility disability [10]. In the current study, application of the multistate modeling extended prior studies by examining incidence of one motor phenotype in relation to another, and their separate and joint associations with the risk of death.

We found that more than 80% of older adults developed mobility disability, based on slowed gait speed, prior to death, a finding that is congruent with prior survey studies reporting

mobility impairment to be the most common type of disability in late-life [29]. In a previous study from the same cohort, we reported that isolated parkinsonian gait impairment was the most common isolated parkinsonian sign in older adults who went on to develop incident parkinsonism [20]. However, a slow gait is a motor phenotype that has a heterogenous group of risk factors including osteoarthritis and peripheral edema that are not risk factors for parkinsonism. In the current study, 60% of participants who developed mobility disability did not develop parkinsonism, a finding that suggests that in a large proportion of older adults mobility disability is not an early sign of parkinsonism. Gait speed may be slowed by numerous defects anywhere in the distributed motor control systems, which extend from the brain through the entire CNS to reach peripheral muscle. This may account in part for why gait speed is a robust, but non-specific early sign of diverse aging motor phenotypes and is not just an early sign of parkinsonism. As such, our data highlight a much more complex relationship between slowed gait speed and parkinsonism in older adults.

The differences observed with transition modeling between the temporal manifestations of mobility disability and parkinsonism were not merely descriptive but were associated with differences in their individual and joint associations with risk of death. These findings have important public health consequences with respect to risk stratification of older adults, emphasizing the importance of vigilance not only for incipient mobility disability, but also for the concurrence of parkinsonian signs. Moreover, the presence of both impairments as harbingers of an increased risk of death might be an important clinical indication for the necessity of more aggressive multi-modal lifestyle interventions and support to maintain survival in older adults.

It is unclear which factors account for the differences in the associations of mobility disability and parkinsonism with the risk of death. Prior studies have shown that more severe motor impairment in older adults is associated with a higher burden of age-related brain pathologies [12, 30] and more severe white matter hyperintensities [31, 32]. Therefore, individuals manifesting impairment of two rather than one motor phenotype may harbor a higher level of brain pathologies disrupting more components of motor networks. Further studies will be needed to elucidate the pathological bases of the findings in this study. Integrating brain imaging and multi-level omics data may help elucidate the underlying mechanisms that account for the patterns of progressive motor impairment observed in the current study.

Since this modeling approach examined nine transitions and six states, there was not enough power to examine additional phenotypes. For example, adding an additional phenotype such as cognitive impairment would increase the number of states from 6 to 17 and the transitions from 9 to 31. Moreover, we have examined the relationship between incident cognitive impairment, mobility disability and death in the current cohort in a prior publication [10]. We therefore excluded individuals with cognitive impairment at baseline to avoid confounding our results. To include additional phenotypes such as cognitive impairment or falls and to identify risk factors for transitions among these aging phenotypes, a larger study would be necessary.

The study has several strengths. Approximately 900 older adults were followed for an average of 10 years with a follow up rate of 90%. Uniform structured data collection was performed at the baseline and follow ups. We used validated instruments to assess two different motor phenotypes to categorize two types of motor impairment rather than using self-reported measures, minimizing recall bias. A novel multistate model was employed to simultaneously examine the onset and inter-relationship of both motor phenotypes as well as their individual and joint relationships to the risk of death.

However, the study has also limitations. Most of the participants were highly educated Caucasians underscoring the need to replicate the study findings in a more general population.

Due to the number of transitions examined in the current study, we excluded participants with cognitive impairment that limits generalization of our study findings to the population with cognitive impairment. While the assessments by nurse clinicians have high inter-rater reliability and short term stability both among nurses and compared to a movement disorders specialist, participants were not assessed by a movement disorders specialist, which might have led to an underestimate of the prevalence of PD or atypical parkinsonism in the current study [18, 19]. Mobility disability was defined by a slow gait speed, rather than using more granular mobility metrics including sway score not collected longitudinally in this cohort, that may be less sensitive in capturing early stages of mobility impairment.

## Supporting information

**S1 Table. Association of vascular risk factors with mobility disability, parkinsonism, and death classified by the motor impairment status before the outcome.**
(DOCX)

**S2 Table. Association of vascular diseases with mobility disability, parkinsonism, and death classified by the motor impairment status before the outcome.**
(DOCX)

**S3 Table. Association of joint pain with mobility disability, parkinsonism, and death classified by the motor impairment status before the outcome.**
(DOCX)

**S4 Table. Main findings of the study after further adjustment for vascular diseases.**
(DOCX)

**S5 Table. Main findings of the study after further adjustment for joint pain.**
(DOCX)

**S6 Table. Main findings of the study after exclusion of 14 participants who were diagnosed to have Parkinson disease during the study.**
(DOCX)

**S7 Table. Main findings of the study after replacing parkinsonism with bradykinetic parkinsonism.**
(DOCX)

**S1 Fig. States and transitions in a multi-state model of incident mobility disability and risk of death.** This figure describes a multi-state model of incident mobility disability and death. The boxes show the 3 possible states and the arrows show the 3 possible transitions.
(TIF)

**S2 Fig. States and transitions in a multi-state model of incident mobility disability, incident parkinsonism, and risk of death after exclusion of 14 participants with a diagnosis of Parkinson disease.** This figure illustrates the frequencies of participants in each of the six states and nine transitions examined in a multi-state model of incident mobility disability, bradykinetic parkinsonism, and death. Mobility disability was defined as gait speed less than 0.55 m/s in an 8-feet walk test. Parkinsonism was defined by presence of at least two of the four parkinsonian signs (bradykinesia, rigidity, tremor, parkinsonian gait). All participants included in this study were initially without mobility disability or parkinsonism at the analytic baseline. During the course of this study, nine paths for transition were possible between the baseline state of no motor impairment, four intermediate states of varying degrees of motor

impairment, and the final absorbing state of death.
(TIF)

**S3 Fig. States and transitions in a multi-state model of incident mobility disability, incident bradykinetic parkinsonism, and risk of death.** This figure illustrates the frequencies of participants in each of the six states and nine transitions examined in a multi-state model of incident mobility disability, bradykinetic parkinsonism, and death. Mobility disability was defined as gait speed less than 0.55 m/s in an 8-feet walk test. Bradykinetic parkinsonism was defined as presence of bradykinesia and at least one other parkinsonian sign (which are rigidity, tremor, parkinsonian gait). All participants included in this study were initially without mobility disability or parkinsonism at the analytic baseline. During the course of this study, nine paths for transition were possible between the baseline state of no motor impairment, four intermediate states of varying degrees of motor impairment, and the final absorbing state of death.
(TIF)

## Acknowledgments

We thank participants of the Religious Orders Study and the Rush Memory and Aging project. Also, we appreciate Traci Colvin, MPH and Tracey Nowakowski, MA for study coordination, and other staff of the Rush Alzheimer's Disease Center.

## Author Contributions

**Conceptualization:** Shahram Oveisgharan, Aron S. Buchman.

**Formal analysis:** Shahram Oveisgharan, Lei Yu.

**Funding acquisition:** David A. Bennett, Aron S. Buchman.

**Investigation:** Aron S. Buchman.

**Methodology:** Shahram Oveisgharan, Lei Yu, Aron S. Buchman.

**Project administration:** David A. Bennett, Aron S. Buchman.

**Resources:** David A. Bennett.

**Supervision:** Aron S. Buchman.

**Writing – original draft:** Shahram Oveisgharan, Lei Yu, Aron S. Buchman.

**Writing – review & editing:** Shahram Oveisgharan, Lei Yu, David A. Bennett, Aron S. Buchman.

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
