## [Decision Letter · Decision Letter 0]

10 Sep 2020

PONE-D-20-20718

Incident mobility disability, parkinsonism, and mortality in community-dwelling older adults.

PLOS ONE

Dear Dr. Oveisgharan,

Thank you for submitting your manuscript to PLOS ONE. After careful consideration by 3 Reviewers and an Academic Editor, all of the critiques of all three Reviewers must be addressed in detail in a revision to determine publication status. If you are prepared to undertake the work required, I would be pleased to reconsider my decision, but revision of the original submission without directly addressing the critiques of the 3 Reviewers does not guarantee acceptance for publication in PLOS ONE. If the authors do not feel that the queries can be addressed, please consider submitting to another publication medium. A revised submission will be sent out for re-review. The authors are urged to have the manuscript given a hard copyedit for syntax and grammar.

**Comments to the Author**

1. Is the manuscript technically sound, and do the data support the conclusions?

Reviewer #1: Yes

Reviewer #2: Yes

Reviewer #3: Yes

2. Has the statistical analysis been performed appropriately and rigorously? 

Reviewer #1: Yes

Reviewer #2: Yes

Reviewer #3: Yes

3. Have the authors made all data underlying the findings in their manuscript fully available?

Reviewer #1: Yes

Reviewer #2: Yes

Reviewer #3: Yes

4. Is the manuscript presented in an intelligible fashion and written in standard English?

Reviewer #1: Yes

Reviewer #2: Yes

Reviewer #3: Yes

5. Review Comments to the Author

Reviewer #1: This is an interesting and very long (10 years) follow-up study of more than 800 elderly people looking at the onset of Parkinosnism and mobiolity disorder according to clinicalle realistic definitions basen on objective clinical scales/measurements. A mathematical modelling approach (multi state Cox model) was applied to model the transition from no motor impairment to the two single impairments and the two different sequences of both impairments and their relation to the probablity of death. Probably the most interesting finding is the higher probabality to develop Parkinsonism after having developed mobility disorder whereas the probablility to develop mobility diosorder after having devlopped Parkinsonism is not much increased. The increased death probabalility with both mobility disorders and its increase with a combindation of both is not very surprising.

My main concern is the clinical meaning of this study. What do the data tell us? Can we conclude anything for screening or caring for the elderly based on this data?

I would suggest to shorten the disuccusion which in large parts repeats the results and try to allude to these questions if notpossible with the presented data give an outlook on how such analyses could help with this in the future.

Does the finding of the clearly increased riks to davalop Parkinsonism after having devleopped mobility impairment mean that slowing oft gait is an underrecognized early sign of Parkinsonism??

Also table 1 (which is difficult to read, the legend doe not state what is given outside and inside the brackets) could possibly give interesting clinical clues, Could the numbers on other diseases or symptoms give insights into risk factors for the two mobility impairments under study? or were there surprises as some expected rsik factors did not show up here??? Why is that?

Reviewer #2: This manuscript focuses on motor impairment with aging at the population level and association with mortality. Using 2 robust cohorts with long follow-up periods up to 10 years, the authors evaluated the impact of “mobility disability” (aka walking speed) and “parkinsonism” on one another and on risk of death. The manuscript is clear and addresses each of the 4 aims adequately. The main findings are that mobility disability is more common than parkinsonism, that each increases risk of death somewhat (parkinsonism more so than mobility disability) but that having both increases the risk of death even further. Developing parkinsonism does not increase the risk of developing mobility disability, but developing mobility disability does increase the risk of developing parkinsonism. However, the sequence did not impact mortality.

I have some conceptual concerns about the authors employing these two “motor phenotypes.” “Mobility disability” has a clear-cut definition and indicates walking speed. This is a heterogeneous group, as walking speed is often impacted by musculoskeletal issues, peripheral edema and neuropathy, and other conditions that are common with aging. “Parkinsonism” as defined here, is based on UPDRS rating but does not comply with clinical criteria for Parkinson’s disease (bradykinesia plus one of the following- tremor, rigidity, characteristic gait changes) and was determined by trained nurses rather than a movement disorders specialist. It seems possible therefore that participants with other forms of gait disorders that share certain features with PD could be misclassified, and that having either tremor (also very non-specific at this age) or rigidity could then lead to classification as parkinsonism. I understand that the goal of the work was not to diagnose PD, but these distinctions impact how we think about the results in terms of mechanisms and guidance for screening methodologies. To illustrate this, one interpretation of the results is that “mobility disability” represents a prodromal stage of parkinsonism with similar underlying neurobiology, but another interpretation is that as slow gait progresses it can mimic parkinsonian gait. Potential suggestions to address these uncertainties would be

1) sensitivity analysis with “parkinsonism” defined as bradykinesia score > 2 and one other feature,

2) add information if available on whether any participants were diagnosed clinically with PD.

One additional consideration worth mention in the discussion is that if participants were diagnosed clinically with PD, they may have been started on symptomatic medications that would impact their gait speed. If this information is available it would be helpful to include it.

Finally, it is very important to emphasize the participants with cognitive impairment were excluded. As the authors mention, cognition and gait are closely related, and thus the findings of this manuscript may not apply to the general population. The authors should also state how cognitive impairment was defined, to clarify the potential cognitive range of participants that were excluded.

Reviewer #3: Well-structured follow ups are a strength of this study as well as use of a multistate model to simultaneously examine incidence of mobility disability and Parkinsonism and their relationship to risk of death. Use of just gait speed for mobility disability may be one of the limitations of this study as previous studies have shown that longitudinal monitoring of postural sway may yield early detection of progressive motor decline. Measures of postural sway during quiet standing are often used to characterize postural control. (Horak F. B. (2006). Postural orientation and equilibrium: what do we need to know about neural control of balance to prevent falls? Age Ageing 35, 7–11. 10.1093/ageing/afl077)

Other risk factors that could contribute to balance control are specific vestibular deficits, somatosensory and visual deficits that should be taken in to account as risk factors for falls in elderly.

Survival is less in atypical parkinsonian syndromes. Also falls, postural instability are more common in these patients. Where these patients diagnosed by a movement disorders specialist and were you able to further characterize the parkinsonian syndrome? In line with this comment, in line 205-207 you talk about risk of death association in participants who developed mobility disability first or parkinsonism first and that the risk was not different between the two group. Again it is interesting to examine what percentage of these patients had typical vs atypical parkinsonism.

You mentioned that you have excluded the patients who had cognitive impairment at baseline. Did you continue to evaluate cognitive function longitudinally? Previous studies have shown a relationship between worsening of balance and cognitive decline. Day-to-Day Variability of Postural Sway and Its Association With Cognitive Function in Older Adults: A Pilot Study. Julia M. Leach,1,2,3,* Martina Mancini,4 Jeffrey A. Kaye,2,3,5,6 Tamara L. Hayes,2,3 and Fay B. Horak4,6

6. PLOS authors have the option to publish the peer review history of their article (what does this mean?). If published, this will include your full peer review and any attached files.

**Do you want your identity to be public for this peer review?** For information about this choice, including consent withdrawal, please see our Privacy Policy.

Reviewer #1: No

Reviewer #2: No

Reviewer #3: No

We look forward to receiving your revised manuscript.

Kind regards,

Stephen D. Ginsberg, Ph.D.

Section Editor

PLOS ONE

"This work was supported by National Institute of Health grants, R01AG043379,

R01AG047679, R01AG056352, R01AG017917, R01NS078009, P30AG10161, and

R01AG15819."

"NO - The funders had no role in study design, data collection and analysis, decision to publish, or preparation of the manuscript."

---

## [Author Response · Author response to Decision Letter 0]

31 Dec 2020

Dear Dr. Ginsberg

Thank you for the opportunity to revise and resubmit our manuscript. We appreciate the time and expertise of you and the reviewers for the careful reading and helpful comments. A point by point response to each comment is included below. Material changes to the manuscript, including the changes’ locations, are noted below. The referenced pages and lines correspond to the pages and lines of the “Revised Manuscript with Track Changes”. 

Kind regards,

Shahram Oveisgharan, MD

REVIEWER 1 

R1.1 My main concern is the clinical meaning of this study. What do the data tell us? Can we conclude anything for screening or caring for the elderly based on this data?

This is an important point. We hope that the revised discussion contextualizes the meaning of our findings with respect to the clinical care of older adults and directions for further aging research. The added paragraph (Page 20, lines 382-389) is included below:

“The differences observed with transition modeling between the temporal manifestations of mobility disability and parkinsonism were not merely descriptive but were associated with differences in their individual and joint associations with risk of death. These findings have important public health consequences with respect to risk stratification of older adults, emphasizing the importance of vigilance not only for incipient mobility disability, but also for the concurrence of parkinsonian signs. Moreover, the presence of both impairments as harbingers of an increased risk of death might be an important clinical indication for the necessity of more aggressive multi-modal lifestyle interventions and support to maintain survival in older adults.”

R1.2 I would suggest shortening the discussion which in large parts repeats the results and try to allude to these questions if not possible with the presented data give an outlook on how such analyses could help with this in the future.

We have shortened the discussion while adding additional analyses and responding to the concerns raised by the reviewer’s in their comments. 

R1.3 Does the finding of the clearly increased risk to develop Parkinsonism after having developed mobility disability mean that slowing of gait is an under-recognized early sign of Parkinsonism?

The reviewer raised an important point. We found that more than 80% of older adults developed mobility disability, based on slowed gait speed, prior to death, a finding that is congruent with prior survey studies reporting mobility impairment to be the most common type of disability in late-life (29). In a previous study from the same cohort, we reported that isolated parkinsonian gait impairment was the most common isolated parkinsonian sign in older adults who went on to develop incident parkinsonism (20). However, as is mentioned by the reviewer 2 at R2.1, a slow gait is a motor phenotype that has a heterogenous group of risk factors including osteoarthritis and peripheral edema that are not risk factors for parkinsonism. In the current study, 60% of participants who developed mobility disability did not develop parkinsonism, a finding that suggests that in a large proportion of older adults mobility disability is not an early sign of parkinsonism. Gait speed may be slowed by numerous defects anywhere in the distributed motor control systems, which extend from the brain through the entire CNS to reach peripheral muscle. This may account in part for why gait speed is a robust, but non-specific early sign of diverse aging motor phenotypes and is not just an early sign of parkinsonism. As such, our data highlight a much more complex relationship between slowed gait speed and parkinsonism in older adults. This is added to the discussion (Pages 19-20, lines 367-381).

R1.4.A Table 1 is difficult to read, the legend does not state what is given outside and inside the brackets.

We hope the changes in Table 1 clarify what values are included within and outside the brackets. 

R1.4.B Also table 1 could possibly give interesting clinical clues, Could the numbers on other diseases or symptoms give insights into risk factors for the two mobility impairments under study? or were there surprises as some expected risk factors did not show up here??? Why is that?

We thank the reviewer for this comment. In fact, Table 1 data does provide clues for possible risk factors for the two motor phenotypes. For example, we expected joint pain, a proxy for osteoarthritis, to be a risk factor for mobility disability. Table 1 data indicates that 30% of the no motor impairment group (NI) had joint pain at study baseline compared with 37% in the mobility disability (MD) group. Using the multistate modeling (S3 Table), we find that joint pain was a risk factor for mobility disability in the transition from no motor impairment (HR=1.34, 95% CI: 1.13 – 1.57, <0.001). Examining a possible risk factor for parkinsonism, we hypothesized that the number of vascular diseases might be a risk factor for parkinsonism based on our prior studies in which we found cerebrovascular brain pathologies explained more of the late-life parkinsonism progression compared with neurodegenerative brain pathologies (30). Table 1 data indicates that participants of the parkinsonism group (PARK) had more vascular diseases (mean=0.2, SD=0.4) compared with participants in the NI group (mean=0.1, SD=0.3). However, the multistate modeling did not find vascular diseases to be a risk factor for transition to parkinsonism (S2 Table). This may be due in part to the low frequency of vascular disease at study baseline. 

Although the primary objectives of this study were examining relationship between the development of two motor phenotypes and their association with the risk of death, we updated the secondary analyses section of the results (Page 15, lines 265-273) and the supplementary file (S1 – S5 Tables) adding details about the association of the examined risk factors with the motor phenotypes and whether the study’s main findings were attenuated after further adjustment for these risk factors. 

We agree with the reviewer that it is of great importance to identify risk factors associated with each of the two mobility impairments which we examined in this study. To adequately address this important issue, a larger study and a more comprehensive survey of potential risk factors is needed. This point has been added to the discussion. Considering that the primary objective of this manuscript was to examine the relationship between the development of two motor phenotypes and their association with the risk of death, we chose not to pursue this subject in the current study. We hope to address this important question more comprehensively in a future study (Page 22, lines 435-437).

The relevant changes in the revised manuscript are provided below:

Results (Page 15, lines 265-273):

“We examined whether controlling for chronic health conditions attenuated the associations of incident motor impairments and the risk of death (S1 – S3 Tables). Vascular risk factors were not associated with any of the 9 transitions. Vascular diseases increased the risk of death in participants who developed only mobility disability. Joint pain increased the risk of mobility disability in participants without any motor impairment. However, further adjustment for vascular diseases or joint pain did not change the inter-relationship of incident mobility disability and incident parkinsonism or their associations with the risk of death (S4 and S5 Tables).”

Discussion (Page 22, lines 435-437):

“To include additional phenotypes such as cognitive impairment or falls and to identify risk factors for transitions among these aging phenotypes a larger study would be necessary.”

REVIEWER 2

 R2.1 I have some conceptual concerns about the authors employing these two “motor phenotypes.” “Mobility disability” has a clear-cut definition and indicates walking speed. This is a heterogeneous group, as walking speed is often impacted by musculoskeletal issues, peripheral edema and neuropathy, and other conditions that are common with aging. “Parkinsonism” as defined here, is based on UPDRS rating but does not comply with clinical criteria for Parkinson’s disease (bradykinesia plus one of the following- tremor, rigidity, characteristic gait changes) and was determined by trained nurses rather than a movement disorders specialist. It seems possible therefore that participants with other forms of gait disorders that share certain features with PD could be misclassified, and that having either tremor (also very non-specific at this age) or rigidity could then lead to classification as parkinsonism. I understand that the goal of the work was not to diagnose PD, but these distinctions impact how we think about the results in terms of mechanisms and guidance for screening methodologies. To illustrate this, one interpretation of the results is that “mobility disability” represents a prodromal stage of parkinsonism with similar underlying neurobiology, but another interpretation is that as slow gait progresses it can mimic parkinsonian gait. Potential suggestions to address these uncertainties would be a sensitivity analysis with “parkinsonism” defined as bradykinesia score > 2 and one other feature.

In this comment, the reviewer has concerns about both motor phenotypes examined in this study: mobility disability and parkinsonism.

A. Response to the concern about mobility disability:

1) We agree with the reviewer that walking speed is often impacted by musculoskeletal conditions. We examined available joint pain data as a proxy for musculoskeletal conditions. Joint pain at baseline was associated with a higher risk of incident mobility disability, but not with the risk of parkinsonism or death (S3 Table). Notably, adjustment for the joint pain did not change our primary findings about the inter-relationship of incident mobility disability and incident parkinsonism (S5 Table). These results have been added to the revised text and are included below (Results, Page 15, lines 265-273):

“We examined whether controlling for chronic health conditions attenuated the associations of incident motor impairments and the risk of death (S1 – S3 Tables). Vascular risk factors were not associated with any of the 9 transitions. Vascular diseases increased the risk of death in participants who developed only mobility disability. Joint pain increased the risk of mobility disability in participants without any motor impairment. However, further adjustment for vascular diseases or joint pain did not change the inter-relationship of incident mobility disability and incident parkinsonism or their associations with risk of death (S4 and S5 Tables).”

B. Response to the concern about parkinsonism:

1) We agree with the reviewer that the UPDRS, used for the assessment of parkinsonism, is not a quantitative instrument. Nonetheless, this instrument has been used for many years in diverse cohorts and predicts important adverse health outcomes (references 4, 17). As pointed out above (R1.3), the longitudinal findings in the current study extend prior cross-sectional reports and provide evidence suggesting that parkinsonism and mobility disability are related but distinct motor phenotypes (Discussion: Pages 19-20, lines 367-381). 

2) We agree with the reviewer that the lack of assessment by a movement disorder specialist may lead to an underestimate of the prevalence of PD or atypical parkinsonism. This has been added as one of the study’s limitations (Discussion, Page 23, lines 449-453). However, we have used the term “parkinsonism” which may capture a much broader and diverse syndrome as has been done by others including well-known and well-respected movement disorder specialists (2,4).

3) Our study, like other community or population-based studies, does not use a movement disorder specialist, but relies on self-report or medical records for the diagnosis of PD and our studies employ nurse clinicians for the assessment of parkinsonism. In several previous publications, we have reported data showing that the UPDRS assessments by the nurse clinicians and the categorization of incident parkinsonism have high inter-rater reliability and short-term stability both among nurses and compared with a movement disorders specialist (Methods: Page 6, lines 111-115; References 17-18). Our prior postmortem studies of adults with parkinsonism indicate that late-life parkinsonism is a heterogeneous disorder that is most commonly related to cerebrovascular disease pathologies (>80%) while PD pathology was observed in a minority of cases (<10%). Moreover, older adults without a clinical diagnosis of PD but with postmortem evidence of PD pathology or adults with a clinical diagnosis of PD most commonly show mixed-brain pathologies (Reference 30). Thus, both the clinical prevalence of PD in the population and postmortem evidence of PD pathology is low.

4) Nonetheless, to highlight the robustness of our findings, we have added the suggested sensitivity analysis in which incident parkinsonism was defined as the presence of bradykinesia, a cardinal sign of PD, with one or more of the other three parkinsonian signs. Analysis using this new definition for incident parkinsonism did not change our main findings, which are summarized and added to the revised text (Results: Page 16, lines 297-303) and the supplementary material (S3 Fig, S7 Table). 

The changes outlined above which have been added to the revised manuscript to address the reviewer’s concerns are provided below. 

Methods (Page 6, lines 111-115)

“Nurse clinicians assessed parkinsonian gait, rigidity, bradykinesia, and tremor annually using 26 items from a modified motor section of the original United Parkinson’s Disease Rating Scale (UPDRS) (17). These measures have high inter-rater reliability and short-term stability among nurses and compared with a movement disorders specialist (18).” 

Results (Page 16, lines 297-303):

“Of the four parkinsonian signs assessed in this study, bradykinesia is considered a cardinal sign for diagnosis of Parkinson disease. Therefore, to examine the robustness of findings, we repeated our analyses after redefining incident parkinsonism based on the presence of bradykinesia together with at least another parkinsonian sign (tremor, rigidity, parkinsonian gait). Our primary findings showing that mobility disability increases the risk of parkinsonism and that individuals with both mobility disability and parkinsonism have the highest risk of death were unchanged (S3 Fig, S7 Table).” 

Discussion (Page 23, lines 449-453):

“While the assessments by nurse clinicians have high inter-rater reliability and short term stability both among nurses and compared to a movement disorders specialist, participants were not assessed by a movement disorders specialist, which might have led to an underestimate of the prevalence of PD or atypical parkinsonism in the current study (18,19).”

R2.2 Add information if available on whether any participants were diagnosed clinically with PD. One additional consideration worth mention in the discussion is that if participants were diagnosed clinically with PD, they may have been started on symptomatic medications that would impact their gait speed. If this information is available it would be helpful to include it.

We repeated our analyses after excluding 14 of 867 participants who had reported Parkinson’s disease during this study and our primary findings were unchanged. This sensitivity analysis is summarized in the revised text (Results: Page 16, lines 290-296) and added to the supplementary material (S2 Fig, S6 Table). 

Results (Page 16, lines 290-296):

“Parkinsonism is a heterogeneous syndrome including older adults receiving different medications, diverse medical conditions and degenerative disorders including PD. To ensure that our findings were not affected by a subset of individuals with PD, we excluded individuals with a clinical diagnosis of PD (n=14) and repeated our analyses. Our primary findings were unchanged: mobility disability was a risk factor for incident parkinsonism and the highest risk of death was among participants with both mobility disability and parkinsonism (S2 Fig, S6 Table).”

R2.2-A Finally, it is very important to emphasize the participants with cognitive impairment were excluded. As the authors mention, cognition and gait are closely related, and thus the findings of this manuscript may not apply to the general population. 

We agree and hope that the changes in the revised manuscript clarify that we excluded individuals with cognitive impairment at baseline from the current analyses. We updated the discussion (Page 22, lines 429-437) by highlighting the facts that examining the relationship between gait speed and cognition was reported in a previous study (Reference 10), and that our sample size for the current study limited our analyses to only 2 intermediate states (mobility disability and parkinsonism). We also have noted that this aspect of the current study design limits generalizing its findings to the general population which includes the full spectrum of late-life cognitive impairment (Discussion: Page 23, lines 447-449). The revisions are provided below: 

Discussion (Page 22, lines 429-437):

“Since this modeling approach examined nine transitions and six states, there was not enough power to examine additional phenotypes. For example, adding an additional phenotype such as cognitive impairment would increase the number of states from 6 to 17 and the transitions from 9 to 31. Moreover, we have examined the relationship between incident cognitive impairment, mobility disability and death in the current cohort in a prior publication (10). We therefore excluded individuals with cognitive impairment at baseline to avoid confounding our results. To include additional phenotypes such as cognitive impairment or falls and to identify risk factors for transitions among these aging phenotypes, a larger study would be necessary.” 

Discussion (Page 23, lines 447-449):

“Due to the number of transitions examined in the current study, we excluded participants with cognitive impairment that limits generalization of our study findings to the population with cognitive impairment.”

R2.2-B The authors should also state how cognitive impairment was defined, to clarify the potential cognitive range of participants that were excluded. 

Details about the battery employed for cognitive testing and assessment of cognitive status as well as details about the number of participants excluded from the study because of cognitive impairment have been added to the revised text. These changes are provided below (Methods: Page 7, lines 132-139):

“The annual structured evaluation included administration of a battery of 17 cognitive tests. A neuropsychologist reviewed the cognitive tests’ results, and a physician with expertise in dementia reviewed annual findings including the summary of the neuropsychologist and classified the cognitive status of participants as no cognitive impairment, mild cognitive impairment, and dementia including probable and possible Alzheimer’s disease, according to established criteria (23). In this study, we excluded participants whose cognitive status at the baseline was mild cognitive impairment (n=895) or dementia (n=212), or the cognitive status could not get determined (n=33).”

REVIEWER 3

R3.1 Use of just gait speed for mobility disability may be one of the limitations of this study as previous studies have shown that longitudinal monitoring of postural sway may yield early detection of progressive motor decline.

The reviewer makes an important point which cannot be addressed in this analytic cohort as longitudinal measures of sway are not collected as part of the annual gait testing. This is noted as a limitation (Discussion: Page 23, lines 457-460), which is provided below:

“Mobility disability was defined by a slow gait speed, rather than using more granular mobility metrics including sway score not collected longitudinally in this cohort, that may be less sensitive in capturing early stages of mobility impairment.”

R3.2 Other risk factors that could contribute to balance control are specific vestibular deficits, somatosensory and visual deficits that should be taken in to account as risk factors for falls in elderly.

As discussed above in response to R3.1, R2.2-A, and R1.4.B, a much larger sample would be needed to examine these important risk factors and to include falls as an intermediate state together with incident parkinsonism and incident mobility disability and death. These details have been added to the revised text and is provided below (Discussion, Page 22, lines 429-437):

“Since this modeling approach examined nine transitions and six states, there was not enough power to examine additional phenotypes. For example, adding an additional phenotype such as cognitive impairment would increase the number of states from 6 to 17 and the transitions from 9 to 31. Moreover, we have examined the relationship between incident cognitive impairment, mobility disability and death in the current cohort in a prior publication (10). We therefore excluded individuals with cognitive impairment at baseline to avoid confounding our results. To include additional phenotypes such as cognitive impairment or falls and to identify risk factors for transitions among these aging phenotypes, a larger study would be necessary.”

R3.3 Survival is less in atypical parkinsonian syndromes. Also falls, postural instability are more common in these patients. Where these patients diagnosed by a movement disorders specialist and were you able to further characterize the parkinsonian syndrome? In line with this comment, in line 205-207 you talk about risk of death association in participants who developed mobility disability first or parkinsonism first and that the risk was not different between the two group. Again it is interesting to examine what percentage of these patients had typical vs atypical parkinsonism.

We agree with the reviewer. As noted above in R2.1, study participants were not examined by a movement disorder specialist. Therefore, we do not have information about typical and atypical parkinsonism. This point is included as one of the study’s limitations (Discussion, Page 23, lines 449-453), which is provided below:

“While the assessments by nurse clinicians have high inter-rater reliability and short term stability both among nurses and compared to a movement disorders specialist, participants were not assessed by a movement disorders specialist, which might have led to an underestimate of the prevalence of PD or atypical parkinsonism in the current study (18,19).” 

R3.4 You mentioned that you have excluded the patients who had cognitive impairment at baseline. Did you continue to evaluate cognitive function longitudinally? Previous studies have shown a relationship between worsening of balance and cognitive decline.

We agree with the reviewer. Please see our response to your prior comments R3.1 and R3.2, and to R2.2-A, and R1.4.B. In summary, we would need a much larger sample to examine incident cognitive or balance impairment in addition to the two motor phenotypes examined in the current analyses. 

JOURNAL REQUIREMENTS

JR1 Please ensure that your manuscript meets PLOS ONE's style requirements, including those for file naming.

We reviewed the journal requirements and accordingly revised the text and supplementary files.

JR2.1 Please remove any funding-related text from the manuscript.

We removed the funding-related text from the manuscript (Page 1, lines 14-16; Page 24, lines 462-464).

JR2.2 Let us know how you would like to update your Funding Statement. Currently, your Funding Statement reads as follows: "NO - The funders had no role in study design, data collection and analysis, decision to publish, or preparation of the manuscript."

After removal of the funding-related text from the manuscript, we reviewed the Funding Information at the submission site to confirm its accuracy. Moreover, we embrace that “the funders had no role in study design, data collection and analysis, decision to publish, or preparation of the manuscript”. 

JR3 We note that you have indicated that data from this study are available upon request. PLOS only allows data to be available upon request if there are legal or ethical restrictions on sharing data publicly.

We responded to this requirement at the cover letter, as was indicated. 

JR4 Please include captions for your Supporting Information files at the end of your manuscript, and update any in-text citations to match accordingly. 

We followed Journal Requirements regarding supplementary materials and included a heading “Supporting information” at the end of the manuscript composing of the names, titles, and captions of supplementary figures and tables. Moreover, we revised the text to match the Supporting information names (e.g., we replaced Supplementary Table e-1 with S1 Table).

---

## [Decision Letter · Decision Letter 1]

15 Jan 2021

Incident mobility disability, parkinsonism, and mortality in community-dwelling older adults.

PONE-D-20-20718R1

Dear Dr. Oveisgharan,

We’re pleased to inform you that your manuscript has been judged scientifically suitable for publication and will be formally accepted for publication once it meets all outstanding technical requirements.

Kind regards,

Stephen D. Ginsberg, Ph.D.

Section Editor

PLOS ONE

**Comments to the Author**

1. If the authors have adequately addressed your comments raised in a previous round of review and you feel that this manuscript is now acceptable for publication, you may indicate that here to bypass the “Comments to the Author” section, enter your conflict of interest statement in the “Confidential to Editor” section, and submit your "Accept" recommendation.

Reviewer #1: All comments have been addressed

Reviewer #2: All comments have been addressed

2. Is the manuscript technically sound, and do the data support the conclusions?

Reviewer #1: Yes

Reviewer #2: (No Response)

3. Has the statistical analysis been performed appropriately and rigorously? 

Reviewer #1: Yes

Reviewer #2: (No Response)

4. Have the authors made all data underlying the findings in their manuscript fully available?

Reviewer #1: Yes

Reviewer #2: (No Response)

5. Is the manuscript presented in an intelligible fashion and written in standard English?

Reviewer #1: Yes

Reviewer #2: (No Response)

6. Review Comments to the Author

Reviewer #1: (No Response)

Reviewer #2: (No Response)

7. PLOS authors have the option to publish the peer review history of their article (what does this mean?). If published, this will include your full peer review and any attached files.

Reviewer #1: No

Reviewer #2: **Yes: **Kara M. Smith MD

---

## [Editor Report · Acceptance letter]

19 Jan 2021

PONE-D-20-20718R1 

Incident mobility disability, parkinsonism, and mortality in community-dwelling older adults. 

Dear Dr. Oveisgharan:

I'm pleased to inform you that your manuscript has been deemed suitable for publication in PLOS ONE. Congratulations! Your manuscript is now with our production department. 

Kind regards, 

on behalf of

Dr. Stephen D. Ginsberg 

Section Editor

PLOS ONE